# A minimalist computational model of slice hippocampal circuitry based on Neuronify for teaching neuroscience

Lucas Longo[1]*, Thiago Lima[1], Maria Clara Bila[1], João Brogin[2], Jean Faber[3]

1 School of Medical Sciences, Federal Fluminense University - UFF, Niterói, Rio de Janeiro, Brazil,
2 Department of Mechanical Engineering, Universidade Estadual de São Paulo - UNESP, São Paulo, Brazil, 3 Department of Neurology and Neurosurgery, Escola Paulista de Medicina, Universidade Federal de São Paulo - UNIFESP, São Paulo, Brazil

* lucaslongo@id.uff.br

## Abstract

The hippocampal formation is vital for processing memory, learning, and spatial navigation. Existing methods are obsolete to address new emerging questions as our understanding of hippocampal circuits and its connections advances. Hence, new techniques with an accessible approach for visualizing and understanding its inner connections and circuitry are needed. Research requires a quick update of textbooks and a better integration of new media to facilitate the teaching of these neural structures. For instance, pictures and diagrams are not enough to fully express the structural and functional effects that each neural circuit imparts. Computational models adapted to these diverse contexts might be a possible solution for such challenge. The construction of minimalist computational models can be an excellent alternative in teaching complex dynamics since they reduce the use of animal models, amplify and simplify structural relationships, promote quick and easy visualization, and uncover possible functional and structural interventions with an educational goal. This interactivity is crucial for a better understanding of the causal relationships between nuclei and neural circuits. Conversely, it is important that those models are simple enough so that any student, regardless of their mathematical background, can understand and manipulate features of interest. Further, software packages that do not require programming knowledge for its use are indispensable, even though this limitation also restricts the representations possible for study. Here, we demonstrate the use of Neuronify software, which uses simple functional representations of neurons and circuits. We represent the most important pathways and connections of the hippocampal formation by building an educational and a simplified models that shows the main known relations between the subregions [Cornu Ammonis (CA)1, CA2, CA3, and CA4], afferent and efferent nucleus (dentate gyrus and subiculum), the first also seeking to couple hippocampal neuroarchitecture, with posterior validation of both by application in an educational context.

**Data availability statement:** The software used for the creation of the model might be downloaded from https://ovilab.net/neuronify/. After acquiring it for free, it will be possible to open the file of the complete hippocampal model described in this paper, which consists of the main data developed, available at the link: https://doi.org/10.6084/m9.figshare.28068563.v1 The simplified hippocampal model can also be downloaded from the link below: https://doi.org/10.6084/m9.figshare.28068602.v1

**Funding:** Coordenação de Aperfeiçoamento de Pessoal de Nível Superior CAPES - code 001. https://www.gov.br/capes/pt-br Conselho Nacional de Desenvolvimento Científico e Tecnológico CNPq/MCT-Instituto Nacional de Neurociência Translacional (INNT): 573604/2008-8. https://innt.org.br/

**Competing interests:** The authors have declared that no competing interests exist.

## Introduction

The field of education in neuroscience faces many specific challenges that differ from other areas of science because of its particularities [1]. Specifically, the requirement of a special set of pedagogic strategies and materials, the limited range of tools available for investigation (e.g., electrophysiological recordings, behavioral index, biochemical markers), and the complex technical language of area of study [2]. The resources required to teach the different aspects of the brain function are limiting due to various factors, e.g., books show a static view of the brain and fail to show the dynamics in a didactic way, and animal models demand technical skill and thorough dedication as they are difficult to manipulate and need to be used responsibly.

In comparison, computational models are an interesting alternative solution since they bypass many issues [3], such as forcing students to learn by simple figures and euthanizing animals, thus supporting initiatives like the 3Rs (replace, reuse and refine) [4]. Computational models offer a fun, easy and interactive way of studying and understanding the complexity of neural circuits and physiological processes.

However, computational models present a series of limitations. The most common is that they generally require knowledge of programming logic [5]. Additionally, since the mathematical descriptions can be too abstract, researchers investigating biological basis use it minimally.

A few software packages exist that propose simple biological representations with user-friendly interfaces, thus allowing anyone to manage and construct their own biological system [6]. Here, we use the free software Neuronify [7], an intuitive platform with a user-friendly interface that does not require expertise in any programming language. Also, many other applications exist that allow the construction of different types of computational models of biological systems, which are not restricted to neuroscience, such as PhET™, a simulator for physics, chemistry and biology [8], "portal SESI educação" from Brazil, MERLOT Biology™, and Atomify™. These simulators have a simple interface and are used for education purposes.

Despite the simple biological representations and some functional limitations offered by Neuronify, we were able to build the complex inner circuitry of hippocampal (HP) formation using only excitatory and inhibitory leaky integrate-and-fire neurons, adjusting them in specific patterns of connectivity to express their real dynamics. For instance, we represented the main pathways of the trisynaptic circuit [9] and highlighted its subfields. By using the virtual sensors provided by the application, we could show the temporal patterns related to the action potential of specific neurons of each HP subfield.

Considering the HP formation neuroarchitecture [10], this model preserved the disposition of semicircular-like complementary structures from dentate gyrus (DG), going through the subfields of Cornu Ammonis (CA3, CA2 and CA1) to the subiculum (SUB), yielding a comprehensive view of the circuit connections [11]. Additionally, we were able to construct a simplified version of the whole HP circuitry by reducing and emphasizing its main nucleus and path connections. Since this circuitry reduction maintains the same physiological activity outputs, it can be an important educational approach to teach how the brain uses redundancies and 'motifs blocks' to process information [12].

In summary, we show how computational models might be useful for teaching neuroscience. We propose an easy way to understand brain networks [13] through the construction of a HP network, its main features and connections, using Neuronify. This work helps the study and understanding of the HP formation, making it easy to observe the response of its parts to stimuli in different conditions determined by the experimenter, simplifying and dynamizing the teaching-learning process.

## Materials and methods

The computational model describes the main circuits of the hippocampus and its correlations considering its input-output information. By a motif-based approach [14] of subregions in HP formation, patterns of recursion, inhibition, and propagation of input signals were established, thus setting the connections between these microcircuits. The motif representation uses the minimum number of neurons (excitatory and inhibitory) and synapses, hence emphasizing its main structural connections and functional relationships. This representation makes the model description more intuitive, and morphologically accurate to a HP slice.

### Neuronify

We used the software Neuronify since it is a user-friendly software with simple features built specifically for educational purposes [15]. The software's interface is a simple "grab and drop" system, where the users only need to select the components they want to display on the screen (Fig 1).

Neurons are simulated in the software using the integrate-and-fire model and are available as either leaky or adaptative. The membrane potential of the former decays over time, hence the name, and the latter reduces its firing rate as it receives sustained input, thus, adapting to repeated stimuli [16]. In order to generate a spike, the neuron membrane potential has to reach a certain threshold, afterwards, it returns to its reset potential, and slowly rises to the resting potential, remaining invariable for a fixed amount of time (the absolute refractory period) (Fig 2). Those parameters can be adjusted in the application for each neuron (Fig 2B and 2C).

Neuronify's model also allows neurons to be determined as either excitatory or inhibitory, both of which can form multiple synapses between each other. In order to measure their state, the application has resources such as voltmeters that display from slight changes in membrane potential to full-on spikes (Fig 2A). Additionally, firing rate plots, spike detectors, and loudspeakers can be used to exhibit, respectively, a step line graph, dots or sounds as the spikes occur. It is worth noting that all available features do not display the exact values of their measurement and are restricted to audio or visual representations. Therefore, Neuronify sensors are limited to displaying qualitative data only.

Synapses are simulated in Neuronify with a current-based model that describes the membrane potential changes according to the synaptic inputs it receives, i.e., a linear summation of all excitatory and/or inhibitory stimuli results in a final membrane potential value. Factors such as the difference between membrane potentials or the synaptic reversal potential are not considered [17], thus, the membrane potential is able to change freely which allows faster and easier computation. Three factors determine synaptic parameters in Neuronify: Maximum Current, Time Constant, and Delay [16], all of which can be adjusted by the user.

Finally, in comparison to other alternatives, we choose Neuronify as the software for our model since it offers a simple drag-and-drop platform that any person with a little training could easily use, while other softwares require a more skilled user, such as NEURON, that demands knowledge on programming using Python language, or GENESIS, that simulates subcellular and biochemical parameters, reproducing more details than necessary, exceeding the objective of a simplified model.

### Motif-based approach and model building

The HP circuit has many redundancies and is essentially constructed by links between smaller circuits called motifs [18]. According to Braganza and Beck [14], a circuit motif is "a conserved anatomical pattern of connections between specific

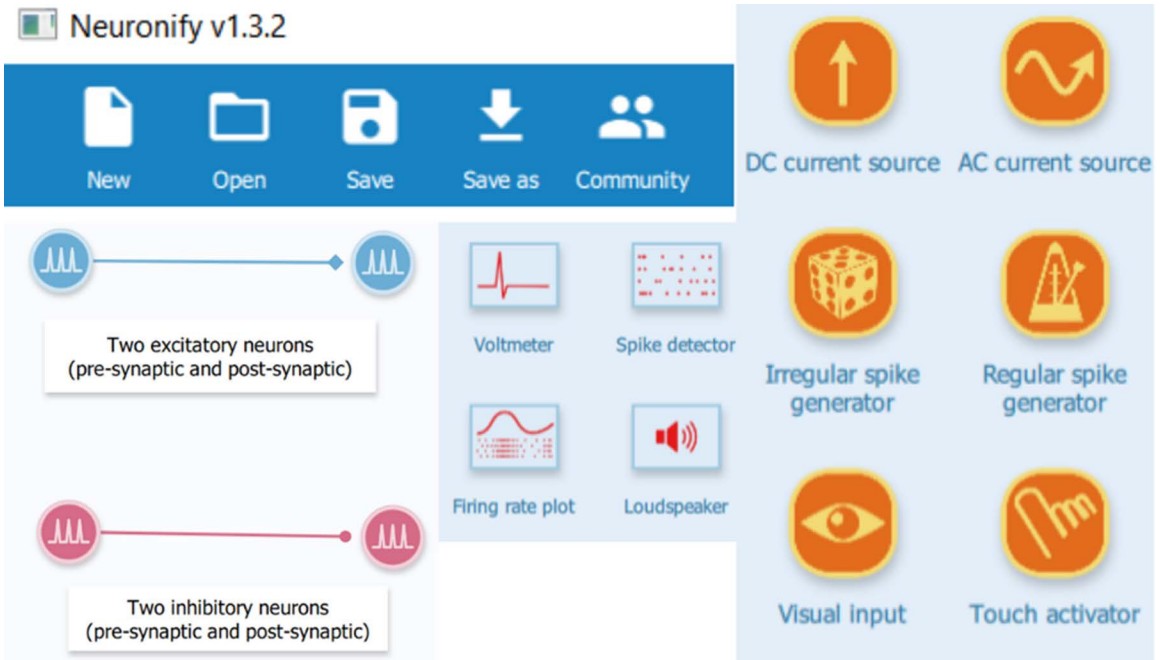

**Fig 1. General portrait of Neuronify main features.** This assembly shows some of the most important components of the software: the option for creating, opening and saving files; the appearance of the neurons, both excitatory and inhibitory, when plotted and connected; the options for sensors that measure the nodes firing by different kinds of plot, such as voltmeter, firing rate plot, spike detector, and a loudspeaker that emits a noise when the selected neuron fires; and the current sources, with a variety of possibilities for stimulating the network by direct or alternating current and generators of spike with regular, irregular or user-defined activation (by using a camera or touching the generator).

cell types." Each motif would work as a computational primitive performing specific operations of information processing such as normalization, and summation [14,18]. Since motifs are the basic blocks of HP circuitry, understanding these blocks is critical in the implementation of any equivalent computational model. Therefore, we chose to build our HP circuit model by assembling pre-determined motifs based on each HP subfield function so as to replicate its spike patterns. The sensors and current sources provided by the software allowed a simultaneous testing process, which made it possible to improve the model during its construction.

We identified and separated HP formation to determine the specific functionalities of each part [19]. The five main features of the subfields can be understood as follows: first, sparsity of DG, i.e., less intensity of output compared to the input; second, the recurrence of stimulation of the CA3 with strong and localized inhibition; third, selectivity of excitatory output by inhibitory feedback of CA2; fourth, the action potentials are feedforwardly transmitted with collateral excitation of CA1; fifth, the almost simultaneous spikes' generation between layers as output of SUB, achieved through simultaneous excitation coupled to inhibitory feedback [20].

The seven motifs were set to match these features. We selected alternative motifs, as follows, that could be incorporated in the network to generate the effects of interest only by using a combination of excitatory and inhibitory neurons (Fig 3):

a) *Simple Feedforward Excitation*: motif relates to the simple and forward propagation of an excitatory signal, causing it to arrive at downstream structures.

b) *Simple Feedback Inhibition*: motif responsible for inhibiting activity on an upstream neuron in a network, preventing it from firing as much as it was previous to excitation of the feedback inhibition network.

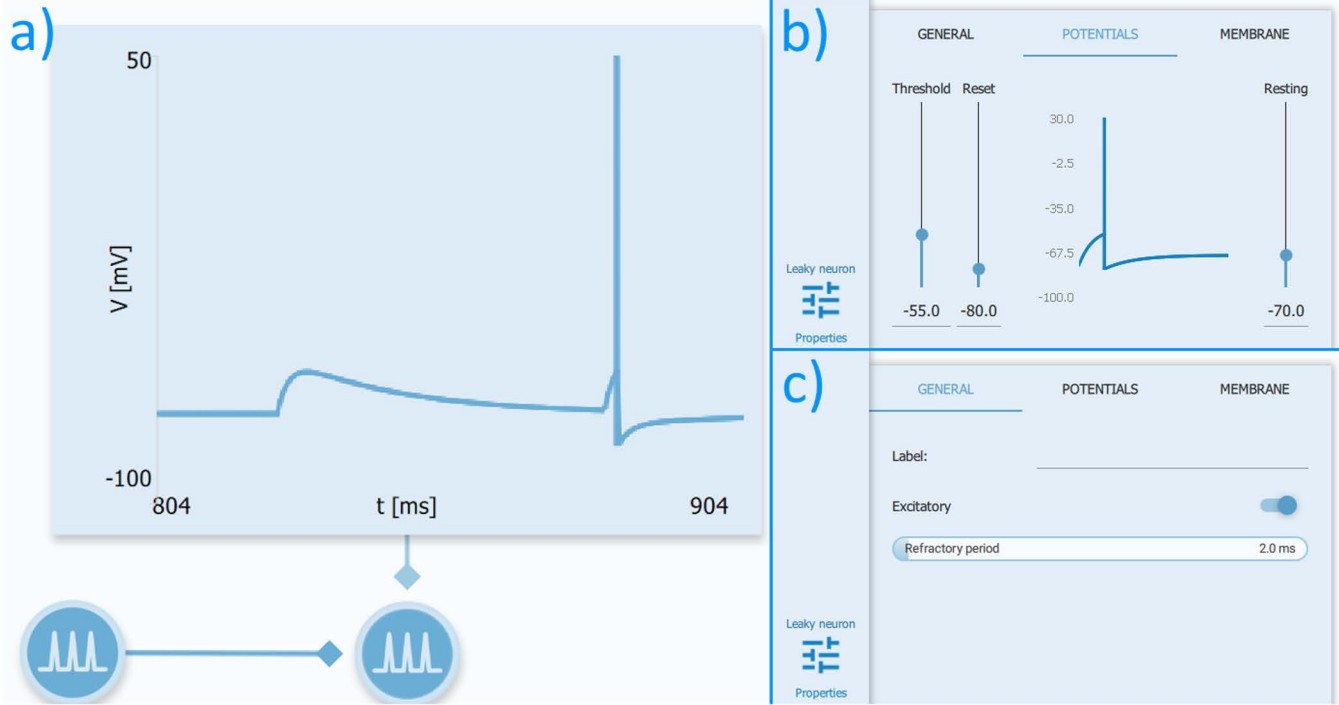

**Fig 2. Intrinsic neuron properties available in Neuronify.** A) Two successive synaptic inputs displayed in a voltmeter. The first input resulted in a membrane potential lower than the threshold potential which failed to generate an action potential and slowly returned to the resting potential. The second input reached the threshold and generated a spike, followed by its immediate reduction to the reset potential and a slow rise until it eventually returns to the resting potential. B) The Threshold, Reset, and Resting potentials, and C) the Refractory Period can be adjusted by the user in the neuron properties panel. It is also possible to adjust the resistance and capacitance of the neuron membrane (not shown).

c) Delayed Feedback Inhibition: similar to 'Simple Feedback Inhibition' where the inhibition occurs on an upstream neuron instead of the same neuron that stimulates the inhibitory cell.

d) *Simple Feedforward Inhibition*: similar to 'Simple Feedback Inhibition' and 'Delayed Feedback Inhibition' where the inhibition occurs on a downstream neuron instead of an upstream neuron.

e) *Simple Feedback Excitation*: a motif characterized by a postsynaptic excitatory neuron that feeds back to the presynaptic one, therefore reinforcing the signal that it receives.

f) *Recurrent Excitation*: similar to '*Simple Feedback Excitation*' where the signal amplified by the recursion between two excitatory interneurons induces a robust representation of a given attribute.

g) *Global Feedback Inhibition*: promotes a selective pattern of activation of the excitatory postsynaptic neurons. An inhibitory neuron selectively chooses which of the neurons are going to fire by varying its synaptic weights while the other neurons remain "silent."

The above motifs are attributed to every structure of the HP formation. DG has a wide presence of simple feedforward excitation and inhibition as well as feedback inhibition. CA3 has feedforward excitation and inhibition, feedback excitation and inhibition, and recurrent excitation. CA2 has feedforward excitation, and simple and global feedback inhibition. CA1 has feedforward excitation, feedback excitation and inhibition, and recurrent excitation. SUB has feedforward excitation and feedback inhibition. Thus, each constructed region shows the peculiarities in its organization similar to a HP slice.

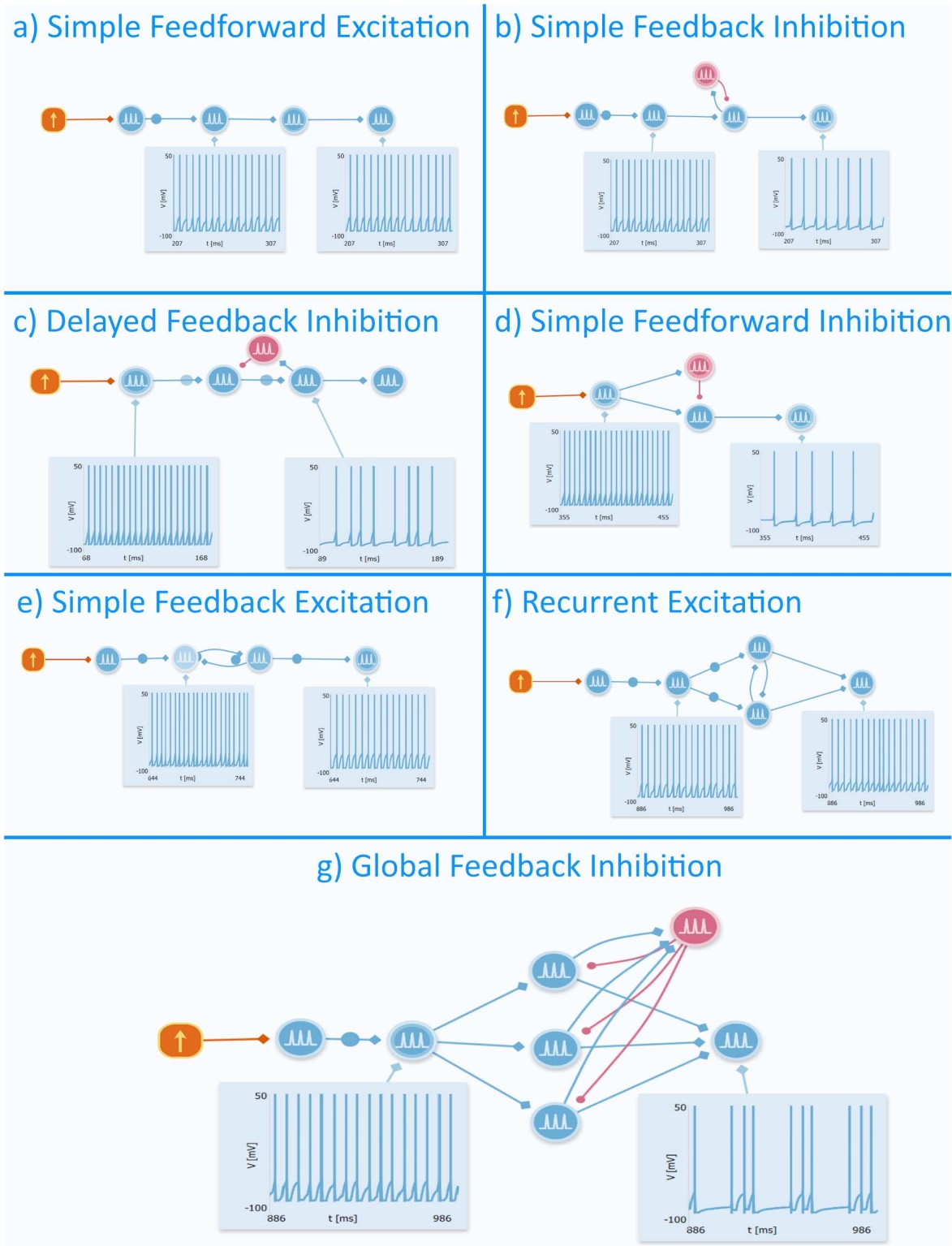

**Fig 3. Circuit motifs and their respective outputs.** This list of constructions shows the motifs needed for building this model as well as their effect on a generic input. These motifs are useful in computational neuroscience since they simplify complex structures by logical pieces with a not really specific anatomic correspondence that is repeated through an area of the nervous system. a) This motif known as "simple feedforward excitation" relates to linear propagation of the signal from one cell to another. Computational mechanism: signal propagation. Computational/ behavioral function: linear

propagation of information. b) This motif known as "simple feedback inhibition" is characterized by inhibitory feedback that limits the maximum excitation of a neuron and may represent a way to control the output of the structure in network. Computational mechanism: output normalization. Computational/ behavioral function: controlling maximum output/ sparsity. c) This motif known as "delayed feedback inhibition" is a variation of the simple feedback inhibition in which the presynaptic neuron is inhibited instead of the neuron that stimulates the inhibitory neuron but resulting in a similar effect as it only limits the presynaptic neuron excitation after it has spiked, guaranteeing information flow. Computational mechanism: output normalization. Computational/ behavioral function: Delayed control of maximum output/ sparsity. d) This motif known as "simple feedforward inhibition" basically plots an inhibitory neuron, which is able to limit the range of activation of another cell by amplifying the possible levels of excitability the next cell will be exposed to. Computational mechanism: input normalization. Computational/ behavioral function: increasing the dynamic range. e) This motif known as "simple feedback excitation" contains a postsynaptic neuron exciting the presynaptic neuron, reinforcing the arriving signal by its repetition between the cells. Computational mechanism: repetition. Computational/ behavioral function: reinforcement of the signal. f) This motif known as "recurrent excitation" is made of excitatory neurons stimulating each other in order to amplify the signal, generating a robust representation of a determined feature. Computational mechanism: amplification. Computational/ behavioral function: robust feature representation. g) This motif known as "global feedback inhibition" presents a characteristic connectivity between its components, with a row of excitatory neurons establishing synapses with an inhibitory cell that feedbacks them, causing an effect called "k-winners take all." It means that the "k" selected cells that receive a less intense inhibitory feedback will fire while the rest of the neurons will stay inactive, being an effective way to discriminate a pattern of interest to a network. Computational mechanism: k-winners take all. Computational/ behavioral function: pattern discrimination.

We emulated the regions on Neuronify using only excitatory and inhibitory neurons. The current sources and sensors were tested during and after each region was composed. We maintained default neuron settings (threshold, resting potential, refractory period, membrane resistance and membrane capacitance) for easy manipulation of the network. Alternatively, we implemented a deficiency of a voltage-dependent sodium channel by changing the membrane's threshold potential [21], while maintaining other properties to equalize every neuron feature. These networks highlighted the spike patterns in relation to the structural connectivity of the cells.

### Hippocampal circuitry models

Using a motif-based approach, we modeled two versions of the HP circuitry in Neuronify: An educational and a simplified model. The former focus on portraying the following three aspects of the HP circuitry: 1) Neuroarchitecture, 2) Motif distribution, and 3) Spike Patterns, whereas the latter lacks on anatomical ressemblance as it prioritizes 'simplicity'. Therefore, both of them have the DG, CA3, CA2, CA1, and SUB subregions which are distinguishable by their motif composition and connected according to the tri-synaptic circuit [22]. Despite that, they diverge on other points, such as the number of neurons and their spatial distribution. That is because the simplified version consists of the minimal number of motifs necessary to emulate specific spike patterns of each subregion.

**Educational HP circuitry model.** The educational model focused on specific motifs and spike patterns determined for each region of the HP based on literature data. The large number of neurons was achieved since only coupling the subregion motifs itself wouldn't be enough for anatomical similarity. Thus, the connections patterns' were maintained in a robust network as described according to morphological resemblance, thereby emphasizing the educational purpose of the network, we created connections similar to that of a histological HP slice (Fig 4).

The medial entorhinal cortex (MEC), subregions II (EC II) and III (EC III), was represented by a row of excitatory neurons composed of six sources of input. This row of neurons received connections from current sources chosen by the researcher. These represent the main pathways by which signals arrive at the HP formation in DG [23]. These six excitatory neurons were designed to project efferent to the DG, CA3 and CA1. A feedforward excitation motif allowed the efferents to receive versatile types of input from the current sources provided by Neuronify.

In DG, the neurons were arranged into a C-shape form with alternating excitatory-inhibitory- excitatory three-layer structure similar to the actual cytoarchitecture of the archicortex [24]. The first layer had a total number of 16 excitatory neurons which was the minimum number of cells sufficient to emulate HP functions, while, creating an easy-to-view network. Experimentally, the number of neurons demonstrated the expected performance without compromising the fidelity of

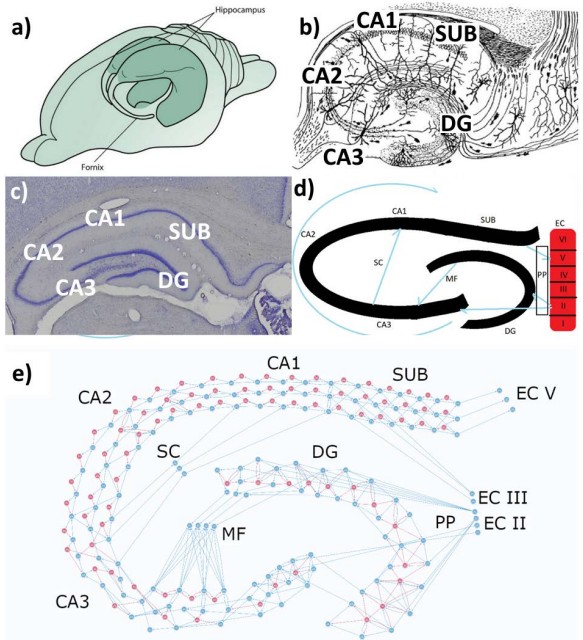

**Fig 4. Hippocampus in human brain, mouse hippocampal slice, schematic hippocampal formation, and the educational HP model.** a) Schematic representation of the location of the two hippocampi inside a rat brain. *Cellular and Molecular Neurophysiology*, fourth edition. Elsevier, 2015. Copyright™. b) Drawing of a transversal slice of the hippocampus by Ramon y Cajal, 1911. c) The illustration simply represents the main structures of hippocampal formation, with the most relevant efferences indicated by the blue arrows. EC II: entorhinal cortex II; EC III: entorhinal cortex III; PP: perforant path; DG: dentate gyrus; MF: mossy fibers; CA1-3: Cornu Ammonis; SC: Schaffer collaterals; SUB: subiculum; EC V: entorhinal cortex V. d) Hippocampal slice in the brain of a wistar rat in a coronal section stained with cresyl violet. (Own source) e) By plotting 227 neurons, 143 excitatory and 84 inhibitory, with a complex machinery of synapses, the neuroarchitecture of hippocampal formation could be preserved whereas its activity could be well demonstrated. Basically, and classically, the information arrives from the EC II and EC III, flows through the "gate," which is the DG by PP up to the CA3 - that receives its afferences mainly by the MF, processes and conducts the information together to the CA2 and CA1 and, in the SUB, shapes the signals directed to neocortex, firstly received in the EC V.

the activity. Similar logic was extended to represent other subregions. The second layer strongly inhibits the others, letting only three neurons, adjacent to the superior limit of the DG representing mossy neurons, without inhibitory input, with 12 inhibitory neurons and a large number of synapses. In turn, the excitatory neurons connect with other excitatory neurons positioned laterally or frontally and at least to one inhibitory neuron that feeds back to them. The last layer consists of 17 excitatory neurons including three mossy neurons. Through the layers, feedforward excitation, feedback inhibition and feedforward inhibition can be observed.

In our model, eight excitatory neurons represent the anatomical input from DG to the CA3 subfield, and the connections between them visually represent the Mossy Fibers. Additionally, we used two neurons in the inferior part of the DG representing input from the SC. For the connections from EC II, we used two neurons directly connected from the PP, which are the main pathway of fibers from the neocortex to HP formation [25].

Further, the subfields' inner connections were randomly plotted, with 12 excitatory neurons and 3 inhibitory ones showing SC stimulation to each other with a strong aspect of inhibition. However, they were organized in a way that allowed any amount of input signals to be filtered by the inhibitory neurons, strategically connected to afferent neurons and excitatory interneurons. This characteristic is better observed in its portion adjacent to DG, almost located in its hilus, where CA4 can be found [26].

 

Another important feature of this part of the hippocampus is the recursion observed between first afferent pyramidal cells in CA3. This recursion corresponds to the reinforcement of arriving information that will be only processed into the hippocampus by the action of biochemical and electrophysiological mechanisms, thus enhancing the learning [27]. To represent this recursion, we used the motifs feedforward excitation and feedback excitation coupled to inhibitory feedback in CA3 to limit the excitability of the signal-receiver cells.

Next, we represent the connections between CA2 and CA3 [28]. We used four afferent cells from CA3 connecting to the surrounding neurons in a pattern characterized by inhibitory feedback that limits the excitability of these cells and the other ones connected to them. Therefore, one of these pyramidal neurons' synapses onto one inhibitory cell and the other three pyramidal neurons synapse onto two inhibitory neurons each. Also, these afferents establish connections to one excitatory neuron to keep the information flowing to the CA2. Computationally, a greater number of connections between both excitatory and inhibitory neurons were required to generate the huge activation attached to a high level of inhibitory feedback in CA2. This structure required a total number of 11 excitatory and 4 inhibitory neurons.

Furthermore, we must consider that CA3's border with CA2 is not precise since these subfields are not anatomically well established [29]. However, a "transition area" that is still a part of CA3 shows a gradual regularization of motifs more related to CA2. This area also has efferents directly to CA1. Assuming the role of SC, we placed three neurons between the area and CA1 to mediate these connections. Hence, the way out CA3 leads information to CA2 by three layers of neurons connected in a pattern of feedforward excitation coupled to feedback inhibition, maintaining the pseudo-unidirectional propagation of information [30]. The boundary of the transition area is immediately after the last (upper) excitatory cell, where the SC is present. Here, SC was modeled by ten excitatory and six inhibitory neurons disposed in connectivity patterns that mix the partially randomly plotted CA3 neurons with great recursion and limitation of excitability paired to the continuous propagation and inhibitory feedback typically seen in CA2.

Moreover, CA2 has a feature of global feedback inhibition [31], by which it selectively enhances synapses according to the stimulus provided [32]. In our model, this feature was represented by a feedback system of three neurons, one from each layer to the same inhibitory neurons where synapses might have their weights changed to strengthen few connections over others, thus composing the global feedback inhibition motif.

In CA1, 23 neurons were used to maintain proportional numbers in the subfields of the three-layer structure. This configuration was sufficiently large to conduct experiments even in a non-scientific computer, while also including the pattern of linear propagation feedforward excitation and inhibitory feedback mechanisms.

In our model, CA1 also received input directly from EC III by PP, showing a different pattern of organization. This pattern was represented as a row of excitatory neurons that stimulate itself located before the SUB layer [33]. The configuration was adopted to maximize the stimulation of the network since beyond the SUB layer the information will flow from the HP formation to the neocortex.

In summary, a total number of 15 excitatory and 15 inhibitory neurons were plotted, where 27 of them were designed to keep the feedforward excitation coupled to feedback inhibitions. The last three excitatory neurons denote recurrent connections, thus stimulating each other to enhance the activation of the organized region and start the relative temporal synchronization of the signal.

The SUB structure was designed to be a continuation of the three-layer architecture, but the layers are less independent. Here, the propagation of the signal takes collateral direction and is not linear as seen in CA2 and CA1. Each neuron is configured to stimulate two other neurons in the following row until the end of the SUB, projecting its efferents to the three terminal neurons labeled as entorhinal cortex V (EC V) which is a part of the lateral entorhinal cortex (LEC) [34]. The LEC receives three to five spikes from SUB sequentially and spreads the information to the adjacent neocortex [35]. Considering the configuration of this area, 18 excitatory and 18 inhibitory neurons were plotted maintaining this connectivity pattern of inhibitory feedback and forward and collateral excitation (using the motifs of feedback inhibition and feedforward excitation).

To construct the whole HP circuitry, a total number of 227 (Table 1) excitatory and inhibitory neurons were used and displayed in patterns according to reviewed literature [36]. This construct allowed the analysis of activity of each subfield and component of the HP circuit.

**Simplified HP circuitry model.** In order to highlight the function and importance of each motif in the model, we also built a simplified version of the HP circuitry in Neuronify by following the same path as the educational version, maintaining the same motifs used in each part of HP. Hence, we plotted and connected neurons according to the seven motifs described previously (Fig 3). Simple feedforward excitation motifs are ubiquitously present throughout the model so information can flow from one point to another. It is worth noting that a high degree of overlap between motifs occurs in the model, consequently, neurons are often seen composing multiple motifs. For example, the same inhibitory neuron can take part in both simple feedforward inhibition and simple feedback inhibition motifs by forming synapses with multiple excitatory neurons. Therefore, the motifs were organized into complex networks and integrated uniquely in each subfield to simulate its main features.

The model input was standardized as a sustained current injection in EC to avoid unclear patterns or unexpected irregularities such as aparrent bursts due to a temporary absence of input (Fig S1). To do so, we used the DC current source, one of the input generators available in Neuronify (Fig 1), to stimulate the EC II and EC III, which resulted in a constant firing rate from those neurons. Consequently, both the educational and simplified models focus on the HP motifs and their spike patterns rather than emulating a realistic circuitry of EC itself, which would be out of this work's scope. Therefore, EC input is similar to a constant current stimulating HP, as a train of action potentials; such simplification is needed once one of our goals is to show a constant frequency of stimuli flowing downstream through the Hippocampus as its subregions modify it with their specific motifs.

In this simplified model, the Entorhinal Cortex II and III consists of one neuron each which are the main sources of input to the HP formation. The former projects directly to DG which is modeled with one inhibitory and four excitatory neurons. As EC II input reaches DG, it first stimulates a single excitatory neuron which composes a simple feedback inhibition motif with the subfield's only inhibitory neuron. That direct inhibition leads to a reduced firing rate at the DG first neuron, therefore increasing the region's signal sparsity which is the main feature we chose to represent in DG. Despite that, the motif output still maintains a relatively high spiking frequency. To address this, the following motif is a simple feedforward inhibition which both increases the interspike interval, and transmits the signal to the last DG neuron.

**Table 1. Number of neurons placed per subregion in the simplified and educational hippocampal formation models.**

| Subregion: | Simplified Model | | Educational Model | |
|---|---|---|---|---|
| | Excitatory neurons: | Inhibitory neurons: | Excitatory neurons: | Inhibitory neurons: |
| EC II | 1 | 0 | 4 | 0 |
| EC III | 1 | 0 | 2 | 0 |
| Dentate Gyrus (DG) | 4 | 1 | 37 | 12 |
| Cornu Ammonis 3 (CA3) | 3 | 2 | 37 | 16 |
| Cornu Ammonis 2 (CA2) | 3 | 2 | 24 | 23 |
| Cornu Ammonis 1 (CA1) | 3 | 0 | 15 | 15 |
| Subiculum (SUB) | 4 | 4 | 18 | 18 |
| EC V | 2 | 0 | 3 | 0 |
| Total | 21 | 9 | 143 | 84 |

The table above shows the number of neurons in each model developed in the present study. In the simplified model, 30 neurons were used to explain the function of the motifs in each hippocampal subregion, employing the minimum number of neurons necessary to achieve this purpose. In the educational model, for functional and didactic reasons, and also visual similarity, a total of 227 neurons were used. In the educational model, the neurons of the CA3-CA2 transition area were counted as CA3 neurons.

Since the central neurons of the simple feedforward inhibition motif only receive excitatory input from one source, any inhibition effect over them of same intensity, i.e., same synapse settings, would result in a null motif output. Therefore, the inhibitory synapse of that motif was set at 2ms and 1.5nA, a value low enough to allow signal transmission. Such adjustment is necessary due to the scarcity of excitatory input due in the simplified model. No alterations on the larger model DG were needed as the motif integration provides enough excitatory input to the cells.

The Mossy Fibers are represented by a long axon which is projected from the last DG cell to the first CA3 neuron. Since Neuronify cannot build an autapse, the first CA3 neurons are organized into a recurrent excitation motif. That structure is essential for promoting the transmission and amplification of the DG input to the subsequent cells of the region. In the following three cells, their organization into feedback and feedforward excitation and inhibition motifs promote the reduction of the oscilations sparsity and also constrain its spread. This circuit depends on excitatory and inhibitory local inputs to generate the recurrent oscillations essential for processing and memory consolidation [27].

Note that the last CA3 inhibitory cell, located at the transition from CA3 to CA2, is connected to the neuron network through a delayed feedback inhibition motif instead of the simple feedback inhibition motif. The reason behind this choice is because the delayed motif allows the presynaptic neuron to fire twice before receiving the inhibitory stimulus. Therefore it produces a milder inhibition, allowing a higher spiking frequency.

The CA3 division transmits the signal through a feedforward excitation to the first neurons of CA2 and CA1 [31]. Thus, the signal reaches the CA2 region and spreads with simple feedforward excitation to the next 2 neurons in the model. The occurrence of simple feedback excitation motifs instead of a simple feedforward motif is to represent the CA1 back-projections to CA3 [37,38]. We plotted a chain of simple feedforward excitation motifs from the first CA1 neuron back to CA3 through all CA2 neurons. Those motifs overlap with the CA2-CA1 motifs, and result in the apparent feedback. Signal transmission towards CA1 is affected by a global feedback inhibition motif, which is built with an inhibitory neuron linked reciprocally with 2 excitatory neurons.

The next subregion, CA1, is composed of five neurons which form simple excitatory and delayed feedback inhibitory motifs with neighbouring cells. The first CA1 neuron works as a gateway as it receives stimulus from EC III, CA2, and CA3 via SC. That signal proceeds downstream to the next two neurons that are connected to each other in a recurrent excitation and a delayed feedback inhibition motifs. The former maximizes the stimulus as the neurons will present a higher frequency of spikes for each input while the latter prevents the recurrent excitation from perpetuating a single input. The choice of a delayed inhibition motif for this region is because the delay allows the targeted neuron to fire at least twice before the inhibition which promotes a stronger output. Therefore, both motifs combined enhance SUB stimulation [33].

As the last HP subregion, the subiculum is modeled as two rows of neurons, each of which has two excitatory and one inhibitory neuron. The first excitatory cells of each row are connected to the following neurons of both rows through simple feedforward excitation motifs to transmit the signal downstream. It also induces temporal synchronization as the signal from one row spreads to the next, and both neurons fire simultaneously. Delayed feedback inhibition motifs in the subregion replicate the spike patterns of subicular bursting cells as the inhibitory effect prevents the affected neuron to reach an action potential temporarily. Such effect can be visualized in voltmeters, as SUB excitatory cells fire at least three times in a row followed by a relatively large period of inactivity. It's interesting to note that even non-bursting subicular cells might present a burst-like pattern of activity if stimulated in higher frequencies, even though our model focused on the representation of the population of bursting cells [39]. EC V is the final output of the model and is represented as two excitatory neurons, which receives input from each SUB row.

## Model validation

The model was validated by applying it to a group of biomedical undergraduate students in Brazil, with 25 people answers to a form organized with questions on a Likert scale. Before and after taking a course on neuroscience and the hippocampus, in a "pre-test" and "post-test" model of comparative evaluation, with the form divided in four sections: "Knowledge

of hippocampal circuits"; "Expectations about using the model"; "Learning expectations" and "Expected practical results". Likert's scale analysis were made by Cronbach's Alpha value. It all was also complemented with a question about the validity of this pedagogical approach to be answered with a score between 0 and 10, and an open question for feedback.

## Results

Using this computational model, we propose a simple platform that allows the study of the structural configuration and functional activities of each subfield of the HP formation, and their relationships. By placing the virtual sensors on each subregion, it was possible to observe the interdependence and the type of information processing that each region performs with respect to the other, and relative to the whole HP circuit.

### HP subfield formations and their dynamics

The signal first arrives at the DG of the HP formation. The pattern of eliciting action potentials is sparse and wide-distributed around the structure, because the granular cells (excitatory neurons) are exposed to intense inhibition. Information gets to DG through PP from EC II and EC III and its activity clearly shows fewer spikes in comparison to that of its neighborhood [40](Fig 5).

Since Neuronify cannot build an autapse, the excitatory neurons from DG that project into CA3 receive feedback input from pyramidal neurons of CA3 to simulate an autapse of CA3 pyramidal SC neuron. This motif is based on a learning rule [41] according to the synchronization of spikes of postsynaptic and presynaptic neurons as a regulator for strengthening an associative memory.

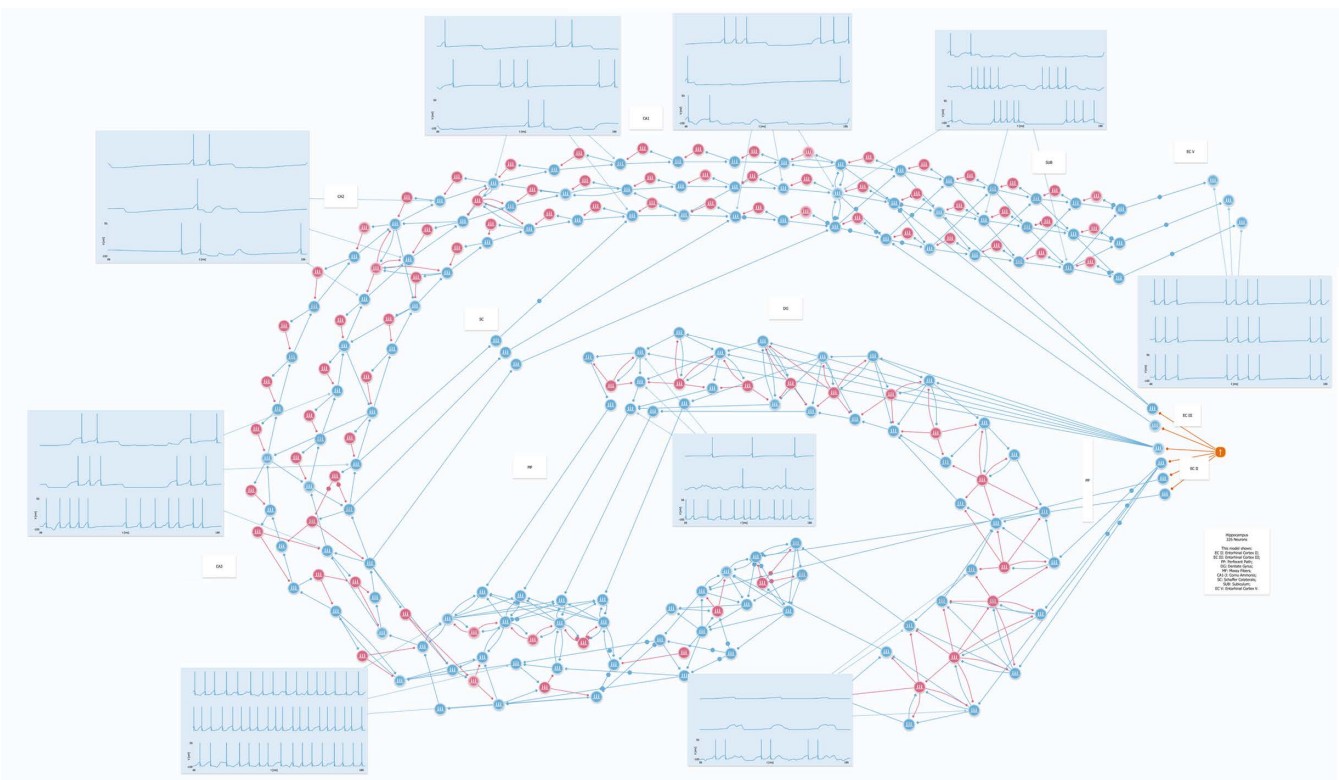

**Fig 5. Portrait of full hippocampal activity.** Wide view of the educational model working. It is possible to visualize the "entire pathway" of the main circuit: information coming from the medial entorhinal cortex (EC) II and EC III and passing through the whole hippocampal formation until it arrives again at the neocortex by EC V.

It is possible to observe recurrent excitation and feedback inhibition (with feedforward excitation, feedback excitation, recurrent excitation, feedforward inhibition and feedback inhibition motifs) between the pyramidal and basket cells in CA3, which is part of the hippocampus proper and located on the fornix [42]. They can be found all over the hippocampus in circuits of feedback inhibition, and hilar interneurons, propagating signals in convergent and divergence patterns. Besides these main components, CA3 receives information directly from EC II and shows two main outputs: CA2, continuing from the fornix, and CA1, through SC [43], composing the well-studied trisynaptic circuit (DG, CA3, CA1, SUB).

CA2 connections followed a basic pattern with three layers of inner pathways representing linear propagation and feedback inhibition [44]. Its pathways have been well characterized recently and data suggests that this structure is important for social memory [45].

The last part of hippocampus proper is CA1, whose circuitry was designed partly in continuity with CA2 and partly mixed with the components of SUB in a more complex configuration of afferents and efferents. The subfield CA1 receives input directly from EC III by PP. It works as an information integrator, being a fundamental part of hippocampus in processing memory, learning, and spatial navigation. Spatial navigation is especially relevant due to the molecular particularities of some of its cells (grid cells and place cells) responsible for mapping the world around us [46].

Despite the widely accepted notion that CA3 connects to CA1, recent studies [37] indicate a bidirectional propagation between CA3 and CA1, where the path from CA3 to CA1 is much more developed. Hence, we plotted only a portion of partial random synapses, so that the signals from CA1 could flow to CA3. Since the literature is unclear whether the recurrent excitation between some neurons of CA1 exists, we plotted it only once.

The output coming from HP formation is forwarded to SUB, whose connections look less random than CA3 but is equally complex, with feedforward propagation applied to excitatory convergence and feedback inhibitory [47]. The normalized output reaches LEC in EC V, spreading itself to the neocortex around.

Not all features of inputs and outputs of hippocampus could be modeled. Yet, the spike activity in each of the subfields by sensors provided by the software match well when compared with biological HP slices [48]. Thus, this simple model aids in the discussion and understanding of structural connections and its associated activities. The difference seen between the model and biological sample is a good opportunity to discuss the universality and redundancies of the neural circuits. Therefore, throughout the process of building the simple model, it is even possible to formulate predictions and hypotheses using solely the structural and functional modifications of the circuit.

### Computational simplification of HP circuitry

The establishment of circuit motifs allowed a subsequent construction of a simpler version of the HP circuitry. We observed a broader view of the flow of information in the structure, thus making it easier to visualize the HP activities [49]. Here, we were able to reproduce most of the complex HP activities and preserve its main recurrent connections, motifs configurations, while using only 31 cells, 21 excitatory and 10 inhibitory (Fig 6).

By turning the complex network into a simpler one, we asked whether it is possible that the brain uses redundancies between neural connections, perhaps for code protection? Reducing the structures and connection rules of HP circuitry can be an interesting educational strategy since it can simplify the understanding of its functional aspects.

This model's simplification shows how effective the motif-approach is. Specially in the context of limited computational power, simplified models like this can be of great applicability depending on the purpose of the experimenter. Additionally, they indicate the effectiveness of neural redundancies since they tend to appear in structures according to their functionalities [50,51].

### Model validation

The results of the Likert scale of the form, evaluated using Cronbach's Alpha, resulted in 0.82 for the pre-test form and a previous value of 0.44 for the post-test. Afterwards, two questions identified as incoherent due to ambiguous wording and

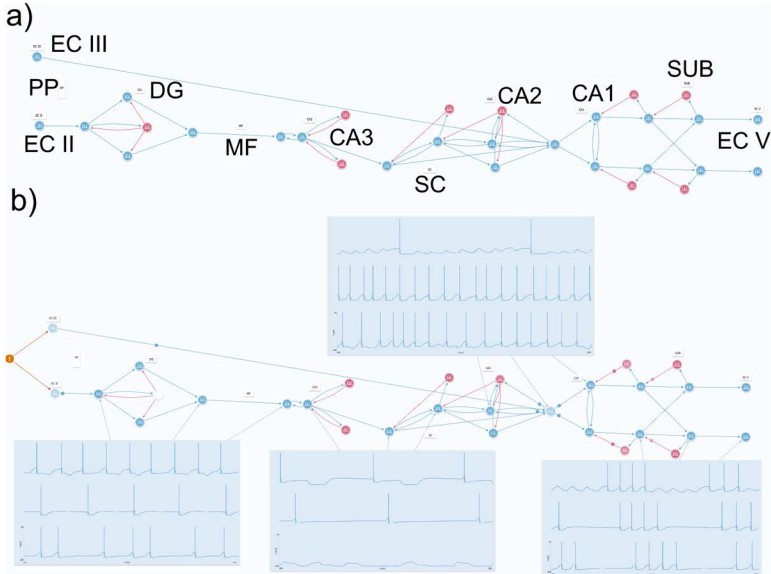

**Fig 6. Simplification of the model of hippocampal circuitry** a) With 31 neurons, it was possible to emulate the main activity of each subregion of hippocampal formation by this motif-driven model (by comparing these recordings with the ones in Fig 5 it is possible to find a good compatibility level). b) The activity of each subregion and the reason for these specific firing patterns are described in the article.

an excessively homogeneous response pattern were adjusted in post-test form, resulting in a new Cronbach's Alpha of 0.57, with an initially unacceptable result becoming moderately reliable, while the pre-test indicated a good reliability. On the question of grading this approach, in the pre-test form, an average score of 9.22 was recorded, while in the post-test the average score was 9.99, indicating that there was an excellent initial perception and even better evaluation of this didactic strategy after its practice.

The students' subjective perception of the model also indicated their approval of it, with positive comments about its didactics, the visualization of the structures and the way the circuitry works. Other suggestions included a request for similar models for other brain circuits and for more in-depth analysis of firing data from individual neurons.

## Discussion

Despite the simplification of the proposed model, it demonstrates high accuracy related to the neuroarchitecture and functional correspondence with real HP circuitry (Fig 7). We believe this type of model and modeling practice can be very useful for teaching and learning neuroscience, such as studying the HP formation, connectivity between neural networks, and electrophysiological properties of neuronal cells.

The simplicity of the platform and biological structures provides anyone a highly accessible way to study this model using basic computers, tablets or even cell phones. Also, it overcomes the limitation of books that fail to demonstrate the cell dynamics.

Computational models like the one proposed here are an interesting alternative for teachers and students interested in understanding the function and structure of the hippocampus or any other brain region. The teaching and learning dynamics open many possibilities not offered by traditional lessons. The model allows for interacting with each structure, manipulating it with different input sources, measuring sensors, changing electrical properties of cells or even varying the position and connectivity of neurons in any region of interest, all through a simple screen and easy mechanism without harming any animals.

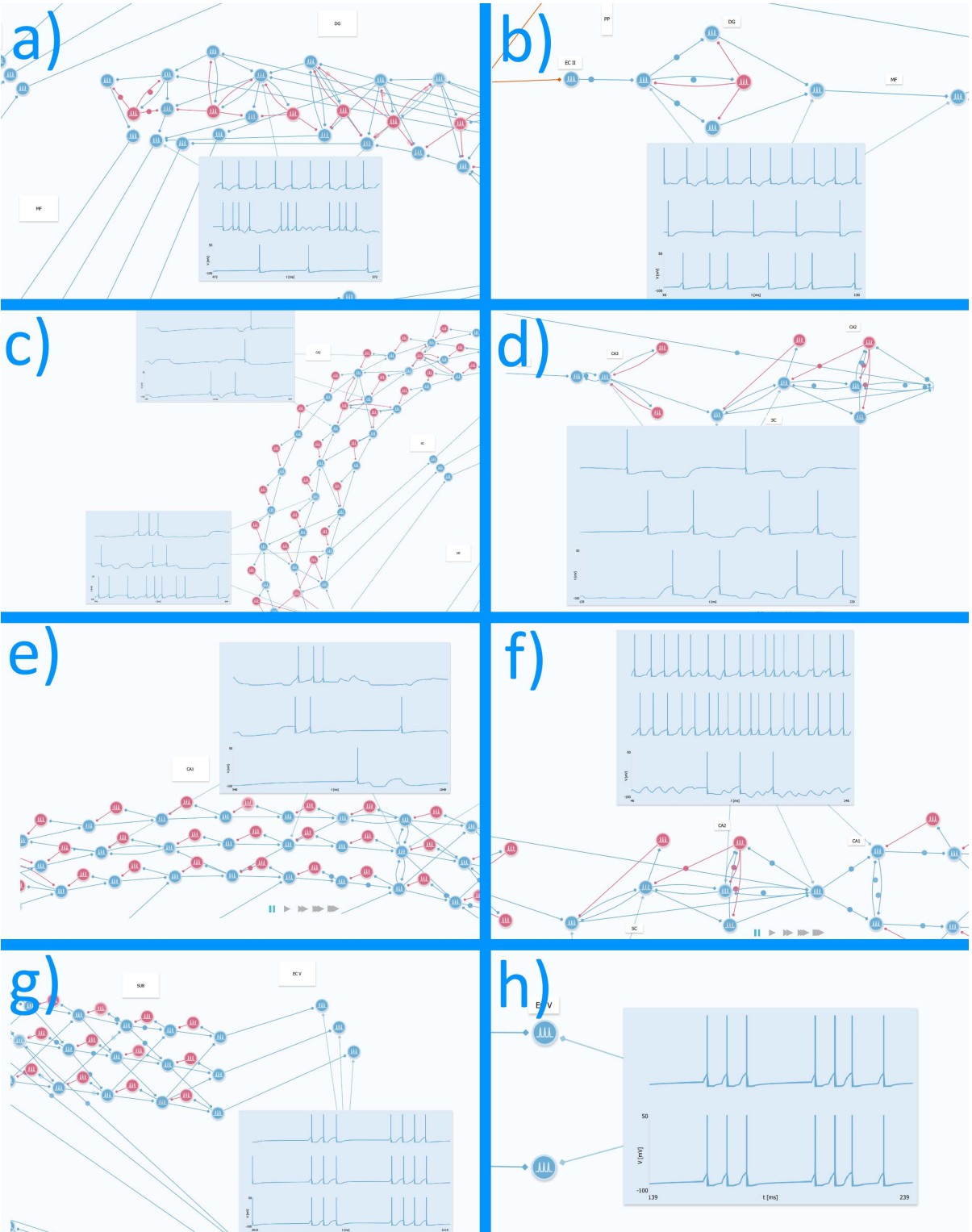

**Fig 7. Comparison between activity of the general model and its simplification.** a) Activity of dentate gyrus of the general model. b) Activity of dentate gyrus of the simplified model. c) Activity of Cornu Ammonis (CA) 3-CA2 of the general model. d) Activity of CA3-CA2 of the simplified model. e) Activity of CA1 of the general model. f) Activity of CA1 of the simplified model. g) Output of subiculum of the general model. h) Output of subiculum of the simplified model.

## Model building

While computational models are useful for education in neuroscience, they have some key limitations. For example, we need to determine how to faithfully represent a realistic and complex HP formation in Neuronify. Despite it being a user-friendly and stable platform, there are limited options to modify the biophysical aspects of cells. To solve this problem, we cluster a variety of cell types into only two categories: (1) every neuron related to a neurotransmitter capable of inducing a depolarization in the cell membrane of a postsynaptic neuron was plotted as excitatory, and (2) every neuron with the opposite effect on postsynaptic membrane potential were designed as inhibitory. Moreover, other neuronal biophysics features including resting potential, membrane capacitance, resistive current were standardized such that discharge patterns were generated via circuit motifs.

Thus, modeling using the circuit motifs approach successfully represents the HP circuitry. The approach allowed the simultaneous prediction of computation effects and construction by only organizing the structure. The choice of the right motifs was crucial, since they determine the functional aspects of each region represented individually and together.

In the case of DG, we implemented the sparsity described in this region. Instead of only minimizing the relative firing rate between the output and the input, we also needed to limit the number of inhibitory cells without creating a large number of synapses between an inhibitory and excitatory neuron. To fix this problem, a proportion of 8:3 was set between the excitatory and inhibitory cells to balance the number of cells and the required properties.

Similarly, in CA3, the challenges included representing the afferent pathways from MF with the strong recurrent connections and coupling these afferent neurons to its inhibitory mechanism. Furthermore, the first signal of action potentials is propagated, processed, and directed to CA2, precisely through the excitatory-inhibitory neural process. To implement this, we defined motifs to properly fit to the dynamics and partially randomize the cell positions. Thus, they could keep their effects even without being restricted to a single and particular organization, but also be able to work in alternative scenarios, e.g., –conditions where the environmental variables could not be fully captured by a simplified model.

The representation of the CA2 subfield addressed selective activation mechanism using the global feedback inhibition motif, which enabled an acceptable level of selectivity that can be virtually modulated by the user. The implementation of this mechanism emphasized the importance of motifs as modular units for building neural circuits.

The CA1 subregion was one of the hardest to implement since it represents different features and specific connections that must be well described. The first portion of CA1 in this model was characterized by the cadence of the signal while the second portion was characterized by a recursion mechanism through which the signal is amplified before arriving at SUB. These characteristics could be implemented because we conjugated the motifs of feedforward excitation and recurrent excitation with feedback inhibition.

Lastly, the SUB region's biggest challenge was to implement the related synchronization of action potentials, so they could represent constructive interference patterns. The solution was to couple the simple feedforward excitation linearly and collaterally to the simple feedback inhibition motif. Thus, the neurons could start firing simultaneously, triggering three to five action potentials per stimuli as described by the literature [52,53].

## Model variants

Once model building relied on a few principles, it is possible to adapt it to represent other HP features or specific circuitry (e.g., Backprojections, or pathogenic alterations). Model variants can be produced as long as their features can be broken down into one or more of the following categories: Neuroarchitecture, motif organization, and spike patterns. For example, the CA3 backprojection to DG is applicable to the model as its features can be represented by all three categories.

Scharfman [54] describes that most of the backprojections to DG hilar neurons come from CA3c where many neurons present a pattern of burst firing. The first step to adapt the model is to replicate that spike pattern, as all CA3 excitatory neurons have a high spiking frequency in its current version (Fig 8A). With the aim of doing so, it is necessary to reduce the number of feedback excitation motifs in the region, as they amplify their input and produce the high firing rates observed. They must be replaced with delayed feedback inhibition motifs, as the delayed inhibition allows the targeted

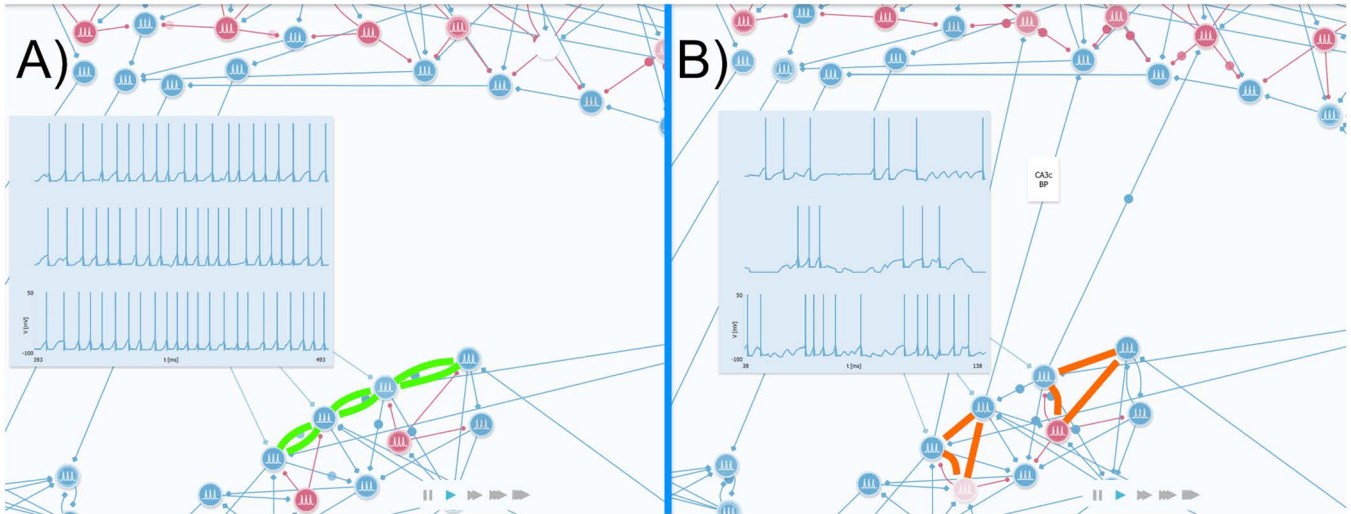

**Fig 8. Model variation including CA3c backprojections compared with the current model The figure on the left A) shows the CA3 in the current version of the educational model.** The neurons arranged into recurrent excitation motifs (green) are responsible for the high firing frequency measured by the voltmeters. Whereas the other figure B) shows a model variant including the CA3c backprojection to DG. An intense inhibition, mainly caused by a sequence of delayed feedback inhibition motifs (in orange) produce a bursting pattern in CA3c cells as shown by the voltmerer. Those cells project back to DG, stimulating two inhibitory and one excitatory cell.

neuron to reach a few action potentials before being silenced (Fig 8B). Due to the low number of neurons present in the area, the addition of that motif will also result in the appearance of both feedforward and simple feedback inhibition motifs. A general increase in CA3c local inhibition may also result in the bursting pattern, although a sequence of delayed feedback inhibition motifs as in SUB is preferable as it produces a more distinguishable burst pattern.

Thereafter, the described connectivity must be applied to the bursting CA3c neurons. The described projections spread through hilar neurons connecting to both mossy cells, and GABAergic interneurons. Since the backprojection effect mainly inhibits granule cells, two of the three CA3 backprojections were plotted targeting DG inhibitory neurons (Fig 8B).

The same approach can be used to apply any other neural structure or hypotheses to the model as long as they are able to fit in the proposed categories.

## Educational effectiveness

As an educational tool, our model represents HP features through two main categories: 1) HP neuroarchitecture, and 2) Motifs and spike patterns. This intends to be a didactic simplification to aid students as they delve into the complex world of neuroscience.

Knowledge of Hippocampal morphology is essential in the study of this area. That is, if a student is able to visualize the dentate gyrus as it partially wraps CA3, the possibility of an axon projection from the former to the latter becomes intuitive due to their physical proximity, and can be easily understood. Thus, fundamental concepts such as the trisynaptic circuit and, in a first moment, its unidirectionality become within the student's grasp. Therefore, neuroanatomy and region connectivity are major features represented in the proposed model, which is why its general appearance resembles a histological HP slice. Each subregion is a cluster of several excitatory and inhibitory neurons along with their axons, it is also possible to identify the mossy fibers projecting from DG to CA3, the SC from CA3 towards CA1, and even a small chain of CA1 backprojections towards CA3 (Fig 5). That way, a student can develop familiarity with the general shape of the HP and its subregions connectivity while interacting with the model.

The HP subregions are represented by an unique collection of neurons organized into specific motifs. Each of which affects its input uniquely, and can be integrated in an area to simulate specific functions or spike patterns. For example, the feedforward excitation motifs are the means of information to go from point A to point B, so their orientation defines the input and output of an area. Other motifs such as the recurrent excitation motif amplify incoming signal, and delayed feedback inhibition motifs in sequence result in a bursting spike pattern. Therefore, the motif-based approach proves valuable in educational contexts because since motifs tie together both morphology and spike patterns, their use enables a broad understanding of the structural substrate behind neuronal organization up to the electrophysiological results stemming from these connections. In this way, it makes it possible to model and understand neuronal networks in a simplified and complete way, with small changes in synapse patterns leading to significant changes in outputs.

The applicability of simplified circuits based on motifs for representing neural networks can be seen in the context of undergraduate courses. For example, where students with little prior knowledge of neuroscience can be exposed to interactive and dynamic models of the circuits they will need to understand, based on neuroarchitecture and electrophysiology comprehension as main goals. In this way, and with the use a user-friendly software such as Neuronify, it is possible that in addition to passively visualizing, these students can effectively engage with the model, editing inputs, intrinsic characteristics and connections of neurons, obtaining a valuable opportunity for active learning of this subject.

## Limitations

Considering all these points, one of the main limitations of this model is its physiological simplicity. Physiological details such as cell types and specific potentials of each neuron could not be fully addressed mainly due to the features of Neuronify. Nevertheless, this simplicity can be seen as an interesting feature, since the model is more intuitive and easier to manipulate during the learning process. Another limitation to consider is the output measurements. The sensors only allow individual neuron measures, not neuronal populations. Particularly it measures only extracellular spike-train activities and does not allow local field potential measurements [55]. Additionally, the stimulation of the network is limited to six options provided by the software that act directly on a selected neuron, without being able to cause an extracellular-like stimulation that simultaneously excites a group of neurons next to an electrode.

Thus, modeling using the circuit motifs approach successfully represents the HP circuitry. The approach allowed the simultaneous prediction of computation effects and construction by only organizing the structure. The choice of the right motifs was crucial, since they determine the functional aspects of each region represented individually and together, but it's important to address its approach limitations, which can be perceived mainly by the impossibility of reliably representing the thousands of synapses to which a single neuron can be subjected, since this complexity of connections would cause the loss of the well-defined firing patterns that are obtained specifically by the topological organization they present. Besides, This demand for spatial organization, coupled with an appreciation for spike patterns as part of this approach in the specific context of the selected software, requires a qualitative assessment of the outputs, without the possibility of quantitative data analysis.

Despite these restrictions, the platform and the model itself present many possibilities for analyzing the signals in each node of the network. We believe this model can be a good object for teaching hippocampus circuitry and its correspondent dynamics. It can even work as an initial step for research on HP disorders based on modifying the connections of the subregions and balancing the functions of each region, although, for this purpose, an interesting future improvement would consist of adapting the software from its free code in order to quantitatively record the firing patterns captured by the sensors, recording frequencies compatible with local field potentials of great use to academic research.

One of the most interesting achievements through this computational model was the demonstration of circuit redundancy, where we were able to reproduce the main aspects of HP circuitry using a motif approach. It provides a good opportunity for students to discuss the arrangement of complex brain networks and how to improve the efficacy of information processing while maintaining energy consumption.

## Conclusions

Using the Neuronify computational platform we were able to implement a model of the hippocampus circuit. The model primarily satisfied the structural architecture encompassing the pathways between the main subregions. Despite the limitations of the platform, such as the use of only excitatory and inhibitory neurons and the restricted biophysical modulation of the cells, we reproduced the main mechanisms of activity modulation using specific circuit blocks. This implementation by block, known as motifs, was fundamental to better understand the structural relations between each subregion and to reproduce their activities.

The model proved to be very robust and at the same time malleable for virtual interventions such as synaptic interruption, short circuit construction, and stimulation. The simplicity of the computation model, along with the user-friendly interface of the platform, makes it important in interactive teaching and learning of neuroscience concepts. This model can potentially contribute to studying other similar dynamics, thus is capable of enriching and assisting the teaching process inside and outside the experimental laboratories.

## Supporting information

**S1 Fig. Educational HP model activity with irregular input.** Wide view of the educational model working with an irregular spike generator stimulating the EC II and EC III neurons. Altough most of the other spike patterns are similar to Fig 5, note the presence of apparent bursts in DG and in the CA3-CA2 transition.
(TIF)

## Author contributions

**Conceptualization:** Jean Faber.

**Methodology:** Lucas Longo, Thiago Lima, Maria Clara Bila.

**Supervision:** Jean Faber.

**Validation:** Thiago Lima, Maria Clara Bila.

**Writing – original draft:** Lucas Longo, João Brogin.

**Writing – review & editing:** Lucas Longo, Thiago Lima, Maria Clara Bila, João Brogin, Jean Faber.

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
