## [Decision Letter · Decision Letter 0]

18 Jul 2024

PONE-D-24-23363A computational model of slice hippocampal circuitry based on Neuronify™ for teaching neurosciencePLOS ONE

Dear Dr. Longo,

Thank you for submitting your manuscript to PLOS ONE. After careful consideration, we feel that it has merit but does not fully meet PLOS ONE’s publication criteria as it currently stands. Therefore, we invite you to submit a revised version of the manuscript that addresses the points raised during the review process. **The reviewers raised relevant points that should be considered by the authors in the revised version of the manuscript.**

We look forward to receiving your revised manuscript.

Kind regards,

Alexandre Hiroaki Kihara, Ph.D.

Academic Editor

PLOS ONE

Journal Requirements:

This work was supported by Conselho Nacional de Desenvolvimento Científico e Tecnológico - CNPq/MCT-Instituto Nacional de Neurociência Translacional (INNT): 573604/2008-8.

4. Thank you for uploading your study's underlying data set. Unfortunately, the repository you have noted in your Data Availability statement does not qualify as an acceptable data repository according to PLOS's standards.

Reviewers' comments:

Reviewer's Responses to Questions

**Comments to the Author**

1. Is the manuscript technically sound, and do the data support the conclusions?

Reviewer #1: Yes

Reviewer #2: Partly

2. Has the statistical analysis been performed appropriately and rigorously? 

Reviewer #1: N/A

Reviewer #2: No

3. Have the authors made all data underlying the findings in their manuscript fully available?

Reviewer #1: Yes

Reviewer #2: No

4. Is the manuscript presented in an intelligible fashion and written in standard English?

Reviewer #1: Yes

Reviewer #2: Yes

5. Review Comments to the Author

**Reviewer #1:**  The present manuscript describes a computational model of the hippocampal circuitry built using the software Neuronify with the intent to teach neuroscience by means of user-friendly software. The software interface looks and works as described, and its features do seem appropriate for students of neuroscience. I thank the authors for their efforts to make neuroscience accessible and understandable to students. However, there are several major issues with the neuroscientific aspects of the work:

1) Both the software and manuscript omit any descriptions of how synaptic dynamics are represented and simulated, and how synaptic parameters are determined; this is a critical aspect of neuronal network function, and from an education perspective the understanding of synaptic function and the differences between different types of synaptic receptors is very important;

2) The manuscript does not describe how the numbers of neurons in the model were determined, and there are no tables with quantitative summaries of the model; the process of determining major structural aspects of the model based on experimental data is something very important that should be taught to students;

3) The input patterns provided to the model are not described anywhere; could the authors describe how the model input patterns are designed and generated? How do the input patterns for this model compare with the known patterns and distributions of EC inputs to the hippocampus?

4) The manuscript often refers to the information passing through the model, and hippocampal circuit function. However, without a definition of the input, and some measures of the output, it is not possible to tell if the model operates according to some hypothetical circuit function. Is it possible to visualize analyze the spike outputs of the model? There is no mention of whether the Neuronify framework itself can analyze spiking activity in some statistical sense, or external tools should be used for that purpose;

5) One of the advantages of computational modeling is that individual components of the neural hierarchy, such as neurons and synapses can be examined, and their properties verified; the present manuscript does not mention how neurons' intrinsic properties, such as f-I curves, are determined, or how ion channel and synaptic mechanisms can be incorporated in the model; however this is a crucial aspect of any network model; it does not become clear how the intrinsic properties of the neurons in the model have been determined;

6) The manuscript claims "From DG, information flows to CA3 through the axons of mossy glutamatergic neurons in MF." -- This is most definitely incorrect. The principal excitatory cells in DG, granule cells, project to the excitatory and some inhibitory neurons in CA3. The granule cell axons are called mossy fibers (MF). The DG mossy cells are thought to project to neurons primarily withing the DG. I do not understand the need for a separate MF neuron type in the model.

7) While I appreciate the idea of a canonical microcircuit for each hippocampal subfield, that then can be used to construct the network structure in a stereotypical fashion, it is not entirely clear whether this is done in a programmatic fashion, or if the constructed manually. What happens if a user desires to use a different (e.g. more detailed) microcircuit motif in one of the subfields? Can they construct the updated network model in an automated fashion, or do they have to recreate all the connections by hand?

8) Related to the previous point, it is unclear how different hypotheses about network structure and function can be evaluated. For instance, it is well known that pyramidal neurons in area CA3c have a significant glutamatergic projection back to DG, where they are thought to connect primarily to mossy cells (The CA3 "backprojection" to the dentate gyrus; Scharfman 2007). Could the authors provide an example how this could be incorporated in a variant of their model? Another useful example would be to show how additional types of interneurons could be incorporated in the model.

Overall, while Neuronify and the hippocampus model developed by the authors look to be a promising tool to teach neuroscience, there are important scientific gaps that need to be addressed before this manuscript is ready for publication.

**Reviewer #2:**  This ambitious study proposes a computational model of hippocampal circuits using Neuronify and represents an innovative approach to neuroscience education. In particular, simplifying and visualizing complex neural circuits using circuit motifs is highly commendable. However, the following points require improvement. If these points are appropriately addressed, we judge that this paper is worthy of acceptance.

1) Effectiveness of the motif-based approach:

a) Please provide a more in-depth discussion of the reasons for selecting motifs in each hippocampus and how they produce specific functions and patterns.

b) Please explain how the motif-based approach specifically contributed to model building and educational effectiveness. Please explain in more detail how the If possible, please provide 1-2 specific educational scenarios to show how understanding the motifs facilitates learning.

c) Please also discuss the limitations and challenges of the motif-based approach and suggest future improvements. For example, please describe how much it captures the complexity of real neural circuits.

2) Regarding reproducibility:

a) Please compare model behavior with actual hippocampal activity quantitatively. b) Please describe setting neuronal parameters (threshold, resting potentials, refractory period, etc.) and model building. Attached is a complete list of parameter settings as supplementary material.

3)Evaluation of Educational Effectiveness:

a) Please provide pilot study data on this model's effectiveness in educational settings.

b) Please provide a tabular comparative analysis with existing educational tools other than Neuronify (e.g., NEURON, Brian, GENESIS) to better demonstrate the uniqueness and superiority of this model. and more clearly demonstrate the uniqueness and superiority of this model.

6. PLOS authors have the option to publish the peer review history of their article (what does this mean? ). If published, this will include your full peer review and any attached files.

**Do you want your identity to be public for this peer review?** For information about this choice, including consent withdrawal, please see our Privacy Policy .

Reviewer #1: **Yes: ** Ivan Georgiev Raikov

Reviewer #2: No

---

## [Author Response · Author response to Decision Letter 0]

4 Jan 2025

Response to Reviewers

Reviewer #1: The present manuscript describes a computational model of the hippocampal circuitry built using the software Neuronify with the intent to teach neuroscience by means of user-friendly software. The software interface looks and works as described, and its features do seem appropriate for students of neuroscience. I thank the authors for their efforts to make neuroscience accessible and understandable to students. However, there are several major issues with the neuroscientific aspects of the work:

The authors thank the reviewer for the consideration and for the meaningful and constructive comments. The Manuscript was revised considering reviewer’s comments and all changes are trackable in the “track changes” version of the Manuscript. Responses are also given bellow.

1) Both the software and manuscript omit any descriptions of how synaptic dynamics are represented and simulated, and how synaptic parameters are determined; this is a critical aspect of neuronal network function, and from an education perspective the understanding of synaptic function and the differences between different types of synaptic receptors is very important;

The authors thank the reviewer for the valuable feedback and the important points that were raised. We recognize that the understanding of these concepts in an educative model is crucial for comprehending neuronal network function. Therefore, we have incorporated an explanation of these points into the Methodology section of the manuscript. We also produced a new image (Figure 1) to summarize the content in a didactic and practical way. Neuronify features integrate-and-fire neurons, available as leaky and adaptive types. The application allows adjustments to some neuron parameters, such as the refractory period, the threshold, reset, and resting membrane potentials, membrane resistance and capacitance. The software allows a qualitative measuremet of the membrane potential of a neuron with a Voltmeter, a spike detector, a loudspeaker or a firing rate plot. Synapses are modeled based on current, linearly summing all excitatory and inhibitory inputs. Key synaptic parameters, such as Maximum Current, Time Constant, and Delay, can be adjusted by the user. The fragment was added in neuronify section of methodology regarding this comment.

2) The manuscript does not describe how the numbers of neurons in the model were determined, and there are no tables with quantitative summaries of the model; the process of determining major structural aspects of the model based on experimental data is something very important that should be taught to students;

Thank you for this comment. We added table 1, which quantifies the inhibitory and excitatory neurons used in each region of the hippocampus in both the educational and simplified versions of the model, the latter was built to highlight the motif’s function. The model was constructed using excitatory and inhibitory neurons in a proportional manner to simulate the activity in each subdivision of the hippocampus from the motifs. Accordingly, 30 neurons were used in the simplified model, which is the minimum amount necessary to explain the impulse pattern and types of motifs in each subfield, and 227 neurons in the educational model, with the objective of facilitating understanding and optimizing student learning due to its morphological resemblance to a HP slice. The distribution by subregion and division into excitatory or inhibitory was shown in the table developed in the article.

3) The input patterns provided to the model are not described anywhere; could the authors describe how the model input patterns are designed and generated? How do the input patterns for this model compare with the known patterns and distributions of EC inputs to the hippocampus?

Thank you for this point. The input patterns are generated by a continuous current that simulates the stimulus's arrival from the EC region. In this context, we prioritized standardizing the input to preserve the functioning of the specific motifs in each region of the model, thereby avoiding interference with the burst patterns of other important areas, such as the SUB. Thus, to achieve this objective, despite understanding the importance of the realism of EC activity, we had to prioritize the maintenance of the hippocampal motifs realism on this concern rather than emulating a realistic circuitry of EC itself, which would be out of this work’s scope. Therefore, EC input is similar to a constant current stimulating HP, as a train of action potentials; such simplification is needed once one of our goals is to show a constant frequency of stimuli flowing downstream through the Hippocampus as its subregions modify it with their specific motifs. Thus, we observe DG sparsity, CA3 signal amplification and SUB bursts, for example, and to clarify this important notation of yours for our future readers, we added this information in the sub-section “Simplified HP Circuitry Model” of “Materials and Methods” of the manuscript.

4) The manuscript often refers to the information passing through the model, and hippocampal circuit function. However, without a definition of the input, and some measures of the output, it is not possible to tell if the model operates according to some hypothetical circuit function. Is it possible to visualize and analyze the spike outputs of the model? There is no mention of whether the Neuronify framework itself can analyze spiking activity in some statistical sense, or external tools should be used for that purpose;

The model allows visualization of the spike outputs, to clarify such possibility can be found in the figure caption 1, where we include the main features that can be used in the application, and in the figure 2A, where we include the step-by-step process of generating the impulse and visualization of the impulse. The spike outputs can be visualized using sensors, such as the voltmeter, firing rate plot and spike detector, which are present in the application. However, the app only allows qualitative analysis, because all the available features are restricted to audio or visual representation, and do not display the precise measurement, not allowing statistical analysis.

5) One of the advantages of computational modeling is that individual components of the neural hierarchy, such as neurons and synapses can be examined, and their properties verified; the present manuscript does not mention how neurons' intrinsic properties, such as f-I curves, are determined, or how ion channel and synaptic mechanisms can be incorporated in the model; however this is a crucial aspect of any network model; it does not become clear how the intrinsic properties of the neurons in the model have been determined;

Thank you for this comment. Neuronify is an application that can represent some qualitative aspects of the impulse, such as spike patterns and fire frequency, which can be modified by adjusting the maximum current, time constant and delay. However, some neuron’s properties can not be visualized in the model, such as f-I curves and the ion channel’s dynamics, because of the limitations of the features that are available in the software. In addition, there are limited options available in the software to modify some of the biophysical aspects of the cells represented in the model. In order to address this problem, we clustered the neurons in two divisions: excitatory neurons, which are represented by every neuron related to a neurotransmitter capable of introducing a depolarization in the membrane of a postsynaptic neuron, and inhibitory neurons, which are the cells that produce the opposite effect on the potential. Moreover, other neuronal biophysics features, such as resting potential and membrane capacitance and resistance current, needed to be standardized such that discharge patterns were generated exclusively via circuit motifs. In order to address this question, we added the figure 2 in the manuscript to make this explanation clearer in the article.

6) The manuscript claims "From DG, information flows to CA3 through the axons of mossy glutamatergic neurons in MF." -- This is most definitely incorrect. The principal excitatory cells in DG, granule cells, project to the excitatory and some inhibitory neurons in CA3. The granule cell axons are called mossy fibers (MF). The DG mossy cells are thought to project to neurons primarily withing the DG. I do not understand the need for a separate MF neuron type in the model.

Thank you for noticing the mistake, and for the explanation and careful attention to the article. We corrected the information, made changes to the model, deleting the neurons corresponding to MF neurons and writing the fragments that could be found in educational effectiveness.

7) While I appreciate the idea of a canonical microcircuit for each hippocampal subfield, that then can be used to construct the network structure in a stereotypical fashion, it is not entirely clear whether this is done in a programmatic fashion, or if the constructed manually. What happens if a user desires to use a different (e.g. more detailed) microcircuit motif in one of the subfields? Can they construct the updated network model in an automated fashion, or do they have to recreate all the connections by hand?

Thank you for your feedback. We would like to clarify that the construction of the model described was carried out manually. If the user desires a different use of the microcircuit design, they have to download the app, and the model and make the alterations by hand, plotting and connecting the neurons to each other. We recognize this is a limitation of the this proposition but, at the same time, it illustrates the versatility of the app and also enrich the understanding of HP circuitry.

8) Related to the previous point, it is unclear how different hypotheses about network structure and function can be evaluated. For instance, it is well known that pyramidal neurons in area CA3c have a significant glutamatergic projection back to DG, where they are thought to connect primarily to mossy cells (The CA3 "backprojection" to the dentate gyrus; Scharfman 2007). Could the authors provide an example how this could be incorporated in a variant of their model? Another useful example would be to show how additional types of interneurons could be incorporated in the model.

The HP model is completely adaptable, since the features represented by it can be grouped into two categories: Spiking patterns, as represented by CA1 bursting cells, DG sparsity, CA3 signal amplification, SUB temporal synchronization, and motifs organization and functionality, as seen throughout the model; and HP neuroarchitecture, as the model mimics the histological organization of a mice HP slice from the DG three cellular layers to the CA1 backprojection to CA3. Those principles allow the model to be adapted, which is an important feature of its intractability and, consequently, to its educational purpose. To address this question properly and emphasize this point we added the section Model Variants in the discussion of the article that elaborates on how to create model variants.

Overall, while Neuronify and the hippocampus model developed by the authors look to be a promising tool to teach neuroscience, there are important scientific gaps that need to be addressed before this manuscript is ready for publication.

We appreciate your considerations and we hope to have addressed the points indicated for modification.

Reviewer #2: This ambitious study proposes a computational model of hippocampal circuits using Neuronify and represents an innovative approach to neuroscience education. In particular, simplifying and visualizing complex neural circuits using circuit motifs is highly commendable. However, the following points require improvement. If these points are appropriately addressed, we judge that this paper is worthy of acceptance.

The authors thank the reviewer for the consideration and for the meaningful and constructive comments. The Manuscript was revised considering reviewer’s comments and changes are trackable in the “track changes” version of the Manuscript. Responses are also given bellow.

1) Effectiveness of the motif-based approach:

a) Please provide a more in-depth discussion of the reasons for selecting motifs in each hippocampus and how they produce specific functions and patterns.

We took your insight in consideration, and elaborated a minute description of each motif role in model building and how they produce specific functions and patterns. Thus, we added a full explanation using the simplified model as a basis to clarify the motifs and their combinations in each region.

b) Please explain how the motif-based approach specifically contributed to model building and educational effectiveness. Please explain in more detail how the If possible, please provide 1-2 specific educational scenarios to show how understanding the motifs facilitates learning.

We appreciate your constructive comments and insert a fragment to the article in the section Educational Effectiveness, which is a new subdivision in the discussion.

c) Please also discuss the limitations and challenges of the motif-based approach and suggest future improvements. For example, please describe how much it captures the complexity of real neural circuits.

We thank you for the analysis, which was fundamental for improving certain points and restructuring the emphasis on specific information. To address this question, we have subdivided the discussion again and included a subsection to address limitations and challenges in the discussion section.

2) Regarding reproducibility:

a) Please compare model behavior with actual hippocampal activity quantitatively.

We regret to inform that we can’t properly answer this question, because of the limitations of the application we can only make a qualitative analysis and compare to actual hippocampal activity, as represent by: neuron spiking frequency reduction and motifs function in each region. More details are present in the limitations section.

b) Please describe setting neuronal parameters (threshold, resting potentials, refractory period, etc.) and model building. Attached is a complete list of parameter settings as supplementary material.

As mentioned in response to comment #1 from the first reviewer, the requested information has been added, and the points have been explained more clearly in the in neuronify section of methodology:

3)Evaluation of Educational Effectiveness:

a) Please provide pilot study data on this model's effectiveness in educational settings.

To address this question we made a pilot study to validate the model. We apply 2 tests to a group of 25 biomedical undergraduate students in Brazil. The first was answered before the students take a course on neuroscience and the hippocampus and the second was applied after for a comparative evaluation using questions on a Likert Scale. We described more details in two fragments: the first was included in the methods and the second in the results.

b) Please provide a tabular comparative analysis with existing educational tools other than Neuronify (e.g., NEURON, Brian, GENESIS) to better demonstrate the uniqueness and superiority of this model. and more clearly demonstrate the uniqueness and superiority of this model.

We couldn’t provide a tabular comparative analysis. However, in order to address this topic, we add the following fragment after compare NEURONIFY, NEURON AND GENESIS. We concluded that Neuronify stands out for being user-friendly and presenting itself in a simple way, allowing the model's objectives to be achieved for educational purposes.

---

## [Decision Letter · Decision Letter 1]

29 Jan 2025

PONE-D-24-23363R1A minimalist computational model of slice hippocampal circuitry based on Neuronify for teaching neurosciencePLOS ONE

Dear Dr. Longo,

Thank you for submitting your manuscript to PLOS ONE. After careful consideration, we feel that it has merit but does not fully meet PLOS ONE’s publication criteria as it currently stands. Therefore, we invite you to submit a revised version of the manuscript that addresses the points raised during the review process. All issues raised by the reviewer should be addressed or replied. Specifically, there are some concerns regarding the figures that must receive authors' attention.

We look forward to receiving your revised manuscript.

Kind regards,

Alexandre Hiroaki Kihara, Ph.D.

Academic Editor

PLOS ONE

Journal Requirements:

Reviewers' comments:

Reviewer's Responses to Questions

**Comments to the Author**

1. If the authors have adequately addressed your comments raised in a previous round of review and you feel that this manuscript is now acceptable for publication, you may indicate that here to bypass the “Comments to the Author” section, enter your conflict of interest statement in the “Confidential to Editor” section, and submit your "Accept" recommendation.

Reviewer #1: All comments have been addressed

2. Is the manuscript technically sound, and do the data support the conclusions?

Reviewer #1: Yes

3. Has the statistical analysis been performed appropriately and rigorously? 

Reviewer #1: N/A

4. Have the authors made all data underlying the findings in their manuscript fully available?

Reviewer #1: Yes

5. Is the manuscript presented in an intelligible fashion and written in standard English?

Reviewer #1: No

6. Review Comments to the Author

Reviewer #1: The revised manuscript is much improved. I thank the authors for their thoughtful revisions. I believe the manuscript is now ready for publication, except for a few minor clarifications, listed below.

1) The text refers to "Cornus Ammonis". The correct term is "Cornu Ammonis".

2) p. 6, the text "fourth, the information cadenced flow with collateral excitation of CA1; fifth, the relative temporal synchronization of spikes as output through simultaneous excitation coupled to inhibitory feedback of SUB". The meaning of this is really unclear, particularly because of the use of non-standard terminology such as the "information cadenced flow". Please revise this so that the meaning is clear to neuroscientists.

3) The order of figures 3,4,5 and the corresponding text describing the models are quite confusing. Consider starting with the current text on microcircuit motifs, then describe the "complete" HP circuit model, then describe the reduced and educational models. Or consider alternative narrative sequences that are more logical than the current text.

4) Figure 4 appears to be truncated in the current manuscript, and it is difficult to understand how it fits in the narrative. Please correct it so that it fits on the page, and consider moving it after the complete model diagram.

5) Figure 5: Panels b) and d) need labels so that they can be related to c) and e). Consider moving panel c) after panel d), as it is conceptually closest to the model.

6) p. 8 "The input patterns are generated by a continuous current that simulates the stimulus's arrival from the EC region. In this context, we prioritized standardizing the input ... one of our goals is to show a constant frequency of stimuli flowing downstream through the Hippocampus as its subregions modify it with their specific..."

- I do not fully understand the meaning of this. Is the intent to have constant input through sustained current injection in order to be able to better observe the transformations of the input through the different HP regions? In that case, please clarify this paragraph and consider adding a supplementary figure that shows what happens when irregular input is provided to the model, as a way to show that more realistic inputs are possible in the model framework.

7) Please clarify in the text what is the model in Figure 5: is this the complete model? Table 1 does not show the numbers for the complete model in Fig. 5. Please modify it so that it mirrors the corresponding narrative in the text.

8) Fig 6 - Fig 11: consider replacing only with Fig. 11, otherwise there are too many figures in the manuscript and it is difficult to follow, especially with the low resolution images. There is redundancy with Fig. 13. Instead of Fig. 6 - Fig. 14, consider including only the current Fig. 11 (complete model), Fig. 12 (reduced model), Fig. 13 (comparison between models), Fig. 14 (CA3 backprojection).

9) Fig. 14 - due to the low image resolution, I am unable to see which neurons have their spiking activity plotted. Does Figure 14B refer to the altered activity of DG granule cells due to the added CA3c backprojection? Please clarify in the figure caption.

7. PLOS authors have the option to publish the peer review history of their article (what does this mean? ). If published, this will include your full peer review and any attached files.

**Do you want your identity to be public for this peer review?** For information about this choice, including consent withdrawal, please see our Privacy Policy .

Reviewer #1: **Yes: ** Ivan Georgiev Raikov

---

## [Author Response · Author response to Decision Letter 1]

4 Feb 2025

Response to Reviewers

Reviewer #1: The revised manuscript is much improved. I thank the authors for their thoughtful revisions. I believe the manuscript is now ready for publication, except for a few minor clarifications, listed below.

R: The authors thank the reviewer for the consideration and for the meaningful and constructive comments. The Manuscript was revised considering reviewer’s comments and changes are trackable in the “track changes” version of the Manuscript. Additionally, at the conclusion of this response, we have compiled a list of the references that were revised in the first revision to achieve the journal requirements.

1) The text refers to "Cornus Ammonis". The correct term is "Cornu Ammonis".

R: We appreciate the correction regarding the term “Cornu Ammonis”. We’ve updated the text accordingly to reflect this accurate terminology.

2) p. 6, the text "fourth, the information cadenced flow with collateral excitation of CA1; fifth, the relative temporal synchronization of spikes as output through simultaneous excitation coupled to inhibitory feedback of SUB". The meaning of this is really unclear, particularly because of the use of non-standard terminology such as the "information cadenced flow". Please revise this so that the meaning is clear to neuroscientists.

R: We thank the reviewer for the important comment on our lexical choices, we seeked the improvement of our writing on this excerpt by adapting the highlighted expressions by others more adequate to the neuroscientific context. The novel composition of this paragraph follows:

“[...] fourth, the action potentials are feedforwardly transmitted with collateral excitation of CA1; fifth, the almost simultaneous spikes' generation between layers as output of SUB, achieved through simultaneous excitation coupled to inhibitory feedback.”

3) The order of figures 3,4,5 and the corresponding text describing the models are quite confusing. Consider starting with the current text on microcircuit motifs, then describe the "complete" HP circuit model, then describe the reduced and educational models. Or consider alternative narrative sequences that are more logical than the current text.

R: We agree that the flow of the description could be improved for better clarity. Based on your suggestion, we have revised the manuscript to enhance the narrative structure. We have now started the materials and methods section by introducing the software (Neuronify), followed by an explanation of the methodology used (motif-based approach and model building). We then describe the models in a new subsection “Hippocampal circuitry models” which begins with the educational (or complete) version and ends explaining the simplified model. Finally, we described the model validation. Also, the educational version of the HP model was referred in the text as both “educational” and “complete”, to avoid confusion, we changed all mentions of “complete” to “educational”. We believe these changes have significantly improved the overall narrative clarity and flow.

4) Figure 4 appears to be truncated in the current manuscript, and it is difficult to understand how it fits in the narrative. Please correct it so that it fits on the page, and consider moving it after the complete model diagram.

R: We appreciate your insightful comment. Upon further consideration, we agree that the inclusion of that figure in its current position does not fit the narrative as it shows a small portion of the simplified model before the whole model is introduced in Figure 5, which is an order that is very likely to cause confusion. In response, we have decided to remove the figure for greater clarity as it is also redudant since the spike patterns it intends to show appear later in the figures 13 and 12.

(The figures 5, 13, and 12 correspond to the figures 4, 6, and 7 in the new version of the manuscript.)

5) Figure 5: Panels b) and d) need labels so that they can be related to c) and e). Consider moving panel c) after panel d), as it is conceptually closest to the model.

R: Thank you for your insightful feedback about our image. We made the required modifications in figure 5.

6) p. 8 "The input patterns are generated by a continuous current that simulates the stimulus's arrival from the EC region. In this context, we prioritized standardizing the input ... one of our goals is to show a constant frequency of stimuli flowing downstream through the Hippocampus as its subregions modify it with their specific..."

- I do not fully understand the meaning of this. Is the intent to have constant input through sustained current injection in order to be able to better observe the transformations of the input through the different HP regions? In that case, please clarify this paragraph and consider adding a supplementary figure that shows what happens when irregular input is provided to the model, as a way to show that more realistic inputs are possible in the model framework.

R: Yes, the intent is indeed to have a constant input so the effect of motifs on the spike patterns can be observed better. We rewrote the paragraph in order to clarify it and also moved that paragraph so it appears before the model description begins. We also added a supporting figure (S1) showing the effect of a random spike generator stimulating the model, aiming to exemplify the importance of constant stimulation for maintaining the characteristics expected by the model as a representation of hippocampal functioning.

7) Please clarify in the text what is the model in Figure 5: is this the complete model? Table 1 does not show the numbers for the complete model in Fig. 5. Please modify it so that it mirrors the corresponding narrative in the text.

R: We appreciate your comment. In the text, we described the completed version as an “educational model”, including in the table, showing the numbers of neurons (excitatory and inhibitory) in each region of the hippocampus simulating the human brain. In response to the comment, we modified the figure capture to make it clearer, and to standardize, we chose to refer to it as the “educational model” in all instances, including in the abstract.

8) Fig 6 - Fig 11: consider replacing only with Fig. 11, otherwise there are too many figures in the manuscript and it is difficult to follow, especially with the low resolution images. There is redundancy with Fig. 13. Instead of Fig. 6 - Fig. 14, consider including only the current Fig. 11 (complete model), Fig. 12 (reduced model), Fig. 13 (comparison between models), Fig. 14 (CA3 backprojection).

R: We appreciate your insightful feedback. In response we deleted figure 6 to figure 10 to reduce the number of figures in the manuscript.

9) Fig. 14 - due to the low image resolution, I am unable to see which neurons have their spiking activity plotted. Does Figure 14B refer to the altered activity of DG granule cells due to the added CA3c backprojection? Please clarify in the figure caption.

R: Thank you for your valuable feedback. We apologize for the low image resolution in Figure 14, which may have made it difficult to properly analyze its content. In response we improved the quality of the figure. The figure 14B shows a voltmeter, which measures the spikes of modified CA3c neurons. We intended to show how the motifs influence the spiking activity, once reducing recurrent excitation and adding inhibitory motifs, the desired CA3c burst pattern is created. We modified the figure capture for clarification.

Journal requirements: Please review your reference list to ensure that it is complete and correct. If you have cited papers that have been retracted, please include the rationale for doing so in the manuscript text, or remove these references and replace them with relevant current references. Any changes to the reference list should be mentioned in the rebuttal letter that accompanies your revised manuscript. If you need to cite a retracted article, indicate the article’s retracted status in the References list and also include a citation and full reference for the retraction notice.

R: To address this request, we carefully revised the reference list and compared the original manuscript with the revised version. In the revised version we used LaTeX to The following modifications were made:

New references added:

[17] Burkitt AN. A Review of the Integrate-and-fire Neuron Model: I. Homogeneous Synaptic Input. Biological Cybernetics. 2006;95(1):1–19. doi:10.1007/s00422-006-0068-6 (included in Neuronify subsection in Materials and Methods)

[22] Amaral DG, Witter MP. The three-dimensional organization of the hippocampal formation: A review of anatomical data. Neuroscience. 1989;31(3):571–591. doi:10.1016/0306-4522(89)90424-7.

[27] Stewart M. Antidromic and orthodromic responses by subicular neurons in rat brain slices. Brain Research. 1997;769(1):71–85. doi:10.1016/S0006-8993(97)00690-2. (included in Simplified HP circuitry model subsection in Materials and Methods)

[52] Scharfman HE. The CA3 “backprojection” to the dentate gyrus. In: Progress in Brain Research. vol. 163. Elsevier; 2007. p. 627–637. Available from: https://linkinghub.elsevier.com/retrieve/pii/S0079612307630349 (included in Model Variants subsection in Discussion)

[25] Sik A, Penttonen M, Ylinen A, Buzsaki G. Hippocampal CA1 interneurons: an in vivo intracellular labeling study. The Journal of Neuroscience. 1995;15(10):6651–6665. doi:10.1523/JNEUROSCI.15-10-06651.1995. (included in Simplified HP circuitry model subsection in Materials and Methods)

References excluded:

-Li, Y.; Mu, Y.; Gage, F. H. Development of Neural Circuits in Adult Hippocampus. Current Topics in Developmental Biology, v. 87, p. 149–174, 2009.

- Giap et al. The Hippocampus Anatomy, Pathophysiology, and Regenerative Capacity. Journal of Head Trauma Rehabilitation, v. 15, n. 3, p. 875–894, 2000.

- Gergues. M. et al. Circuit and molecular architecture of a ventral hippocampal network. Nature Neuroscience, v. 23, p. 1444–1452, 14 set. 2020.

- Latimer, B. et al. Open Source Software Tools for Teaching Neuroscience. The Journal of Undergraduate Neuroscience Education, v. 16, n. 3, p. A197–A202, 3 set. 2018

---

## [Editor Report · Decision Letter 2]

6 Feb 2025

A minimalist computational model of slice hippocampal circuitry based on Neuronify for teaching neuroscience

PONE-D-24-23363R2

Dear Dr. Longo,

We’re pleased to inform you that your manuscript has been judged scientifically suitable for publication and will be formally accepted for publication once it meets all outstanding technical requirements.

Kind regards,

Alexandre Hiroaki Kihara, Ph.D.

Academic Editor

PLOS ONE
---

## [Editor Report · Acceptance letter]

PONE-D-24-23363R2

PLOS ONE

Dear Dr. Longo,

I'm pleased to inform you that your manuscript has been deemed suitable for publication in PLOS ONE. Congratulations! Your manuscript is now being handed over to our production team.

Kind regards,

on behalf of

Dr. Alexandre Hiroaki Kihara

Academic Editor

PLOS ONE